# Fail-safe control of translation initiation by dissociation of eIF2α phosphorylated ternary complexes

**Martin D Jennings[1,2], Christopher J Kershaw[1,2], Tomas Adomavicius[1,2], Graham D Pavitt[1,2]\***

[1]Division of Molecular and Cellular Function, Faculty of Biology Medicine and Health, The University of Manchester, Manchester, United Kingdom; [2]Manchester Academic Health Science Centre, The University of Manchester, Manchester, United Kingdom

**Abstract** Phosphorylation of eIF2α controls translation initiation by restricting the levels of active eIF2-GTP/Met-tRNAi ternary complexes (TC). This modulates the expression of all eukaryotic mRNAs and contributes to the cellular integrated stress response. Key to controlling the activity of eIF2 are translation factors eIF2B and eIF5, thought to primarily function with eIF2-GDP and TC respectively. Using a steady-state kinetics approach with purified proteins we demonstrate that eIF2B binds to eIF2 with equal affinity irrespective of the presence or absence of competing guanine nucleotides. We show that eIF2B can compete with Met-tRNAi for eIF2-GTP and can destabilize TC. When TC is formed with unphosphorylated eIF2, eIF5 can out-compete eIF2B to stabilize TC/eIF5 complexes. However when TC/eIF5 is formed with phosphorylated eIF2, eIF2B outcompetes eIF5 and destabilizes TC. These data uncover competition between eIF2B and eIF5 for TC and identify that phosphorylated eIF2-GTP translation initiation intermediate complexes can be inhibited by eIF2B.

**\*For correspondence:** graham. pavitt@manchester.ac.uk

**Competing interests:** The authors declare that no competing interests exist.

## Introduction

In eukaryotic translation initiation, initiator tRNA (Met-tRNA$_i$) recognises AUG start codons in mRNA as part of a larger complex bound to the 40 S ribosome. Eukaryotic translation initiation factor 2 (eIF2) is the key factor that delivers Met-tRNA$_i$ to ribosomes (*Hinnebusch, 2014*; *Dever et al., 2016*). eIF2 is a G-protein and binds GTP and Met-tRNA$_i$ forming a ternary complex (TC) (*Schmitt et al., 2010*). Together with other initiation factors the TC-bound ribosomal preinitiation complex (PIC) binds near the mRNA 5' cap and scans the mRNA usually to the first AUG codon where Met-tRNAi anticodon/mRNA codon interactions help facilitate translation initiation. Hence TC is critical for delivering tRNA$_i^{Met}$ to 40 S ribosomes, for scanning the 5'UTR and AUG codon recognition (*Hinnebusch, 2014*; *Dever et al., 2016*; *Llácer et al., 2015*). Importantly, in all eukaryotes studied, translation initiation is controlled by regulating the activity of eIF2, see below. Met-tRNA$_i$ has a ~10 fold greater affinity for eIF2-GTP over eIF2-GDP (*Kapp and Lorsch, 2004*). Hence hydrolysis of eIF2-bound GTP and Pi release during AUG codon recognition, facilitates loss of eIF2-GDP from initiating ribosomes (*Algire et al., 2005*). This eIF2-GDP must be converted to an active GTP-bound form for continued active translation initiation. Reactivation of eIF2 relies on eIF2B, a multifunctional guanine nucleotide exchange factor (GEF) (*Pavitt, 2005*).

eIF2B GEF activity facilitates GDP release from eIF2-GDP complexes (*Panniers et al., 1988*; *Pavitt et al., 1998*) enabling GTP and Met-tRNAi binding to eIF2. However, the precise mechanism of eIF2B-mediated GEF action is not yet understood (*Mohammad-Qureshi et al., 2008*), see

**eLife digest** All cells sense and react to changes in the world around them. One way that cells react to threats to their health is by switching off genes required for their normal activity and diverting this energy to switching on genes that deal with the specific stress. A protein called eIF2 controls this general response, which is known as the "integrated stress response". The eIF2 protein is switched on when it binds to a molecule of GTP and switched off when it binds to a molecule of GDP; these two molecules are swapped by another protein called eIF2B.

Previous research revealed that the eIF2B protein attaches to eIF2 that is carrying GDP (also known as eIF2-GDP) and decides whether to switch it to eIF2-GTP or not depending on signals from the integrated stress response. However it was not known if eIF2B could also attach to eIF2-GTP or how this might affect the eIF2 protein, the activity of genes and the integrated stress response.

Now, using proteins extracted from baker's yeast as a model, Jennings et al. studied the interactions between eIF2 and eIF2B. The experiments revealed that, under stressful conditions, eIF2B not only triggers the integrated stress response through eIF2-GDP but also halts any active eIF2-GTP not under the control of this response. Jennings et al. suggest that these two processes represent a fail-safe switch that ensures that the integrated stress responses occurs rapidly whenever the cell is stressed.

It is not clear how important this proposed fail-safe switch is for different types of cells, and so further studies will explore this question. In particular, people with mutations in the five genes that encode eIF2B develop a fatal brain disease and have an impaired integrated stress response, and so studies might check to see if some of these mutations affect the fail-safe switch.

*Figure 1A*. One highly conserved form of translational control involves signalling to and activation of eIF2 kinases that each phosphorylate serine 51 of the eIF2α subunit, for example during periods of cell stress (*Pavitt, 2005*) and known widely as the integrated stress response (ISR). In the ISR, the phosphorylated form of eIF2 [eIF2(αP)] binds eIF2B unproductively forming a GEF-inhibited complex that restricts TC levels (*Rowlands et al., 1988*; *Pavitt et al., 1998*) (*Figure 1A*). This causes a general reduction in protein synthesis initiation rates while at the same time activating translation of ISR-responsive mRNAs (*Young and Wek, 2016*). Genetic and biochemical evidence shows that phosphorylated eIF2α binds to a regulatory site on eIF2B formed by the eIF2Bαβδ subcomplex (*Pavitt et al., 1997*; *Krishnamoorthy et al., 2001*; *Kashiwagi et al., 2016b*). In contrast GEF function is provided by eIF2Bε, which interacts with eIF2β and the GDP/GTP-binding eIF2γ subunit (*Gomez and Pavitt, 2000*; *Asano et al., 1999*; *Alone and Dever, 2006*). The γ subunit of eIF2B binds to, and stimulates the GEF action of, eIF2Bε (*Pavitt et al., 1998*; *Jennings et al., 2013*).

A second eIF2 regulatory factor is eIF5. eIF5 primarily functions as a GTPase activating protein (GAP) for eIF2-GTP within the 48S pre-initiation complex (PIC) (*Paulin et al., 2001*; *Algire et al., 2005*). GTP hydrolysis within eIF2-GTP and subsequent inorganic phosphate release are key events that signal AUG start codon recognition by initiating ribosomes (*Algire et al., 2005*). These events facilitate release of eIF2-GDP/eIF5 complexes from the initiation complex, enabling subsequent 60S joining. eIF5 binds inactive eIF2-GDP and active TC with similar affinity ($K_d$ = 23 nM; [*Algire et al., 2005*]) and we found that eIF5 has a second activity with eIF2-GDP that impairs spontaneous release of GDP from eIF2. This GDP-dissociation inhibitor (GDI) function is important during the ISR when eIF2α is phosphorylated and translation is attenuated as GDI prevents eIF2B independent release of GDP and ensures tight translational control (*Jennings and Pavitt, 2010a*, *2010b*; *Jennings et al., 2016*). Under optimal cell growth conditions, when free eIF2B is available, we found that eIF2B (specifically eIF2Bγε) actively displaces eIF5 from eIF2-GDP prior to its action as a GEF (*Jennings et al., 2013*). Thus the combined actions of eIF2B and eIF5 regulate eIF2 nucleotide status (*Figure 1A*).

GEFs typically function to activate their cognate G protein partners by destabilizing the binding of $Mg^{2+}$ and GDP prompting GDP release to a nucleotide free intermediate that allows GTP to bind. To promote nucleotide exchange GEFs have a higher affinity for nucleotide-free forms of their G protein partners than to the nucleotide bound forms (*Bos et al., 2007*). For example, translation elongation factor eEF1A/eEF1B affinity is 0.125 μM, while affinity of eEF1B for the nucleotide-bound

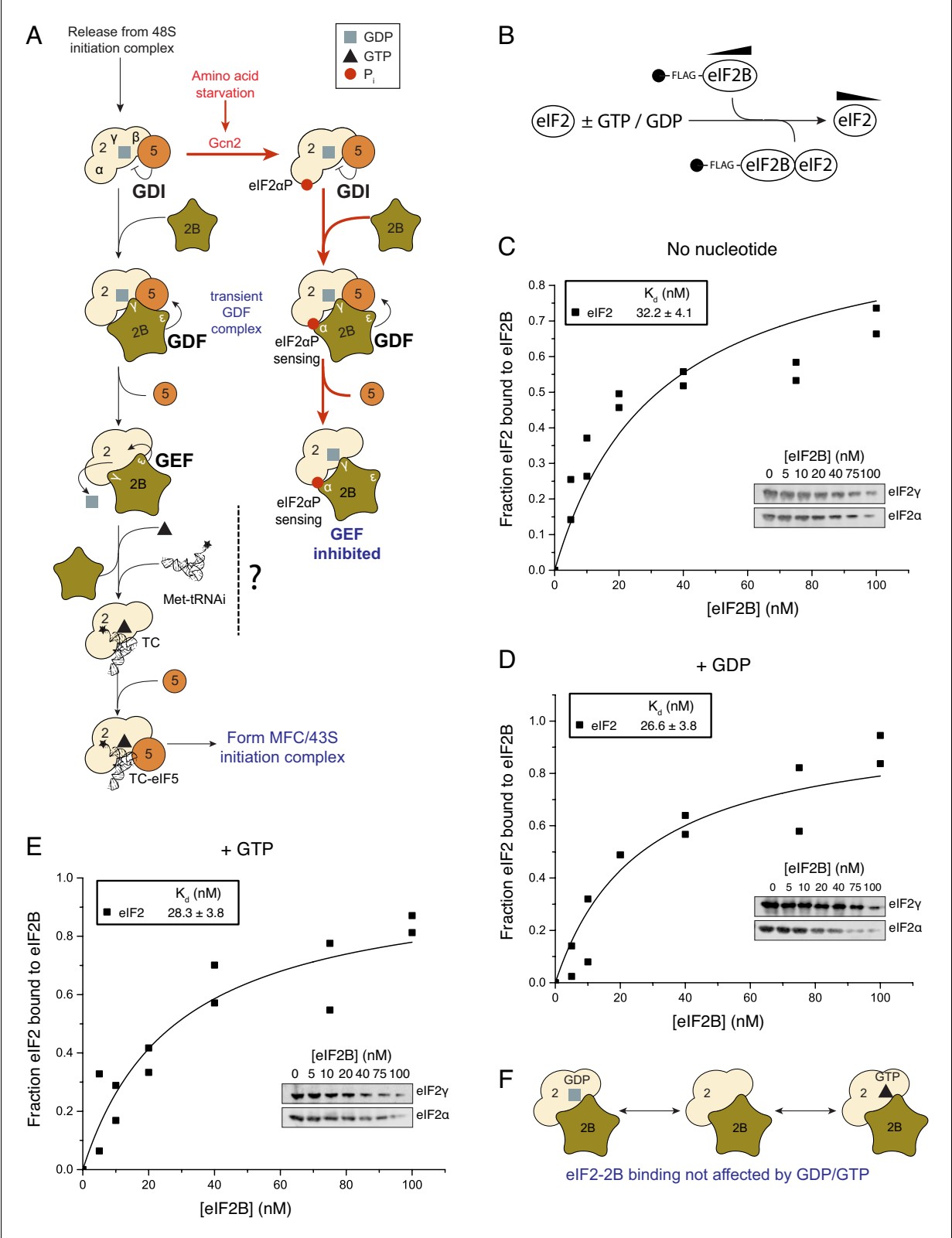

**Figure 1.** eIF2 affinity for eIF2B is unaffected by guanine nucleotides. (**A**) Current model for eIF2 activation and inhibition by phosphorylated eIF2. Interactions and activities are explained in the introduction. (**B**). Overview of eIF2/eIF2B equilibrium binding assay. (**C–E**). Affinity determined by mixing 2 nM eIF2 with increasing concentrations of eIF2B immobilised on anti-Flag resin. Flag resin was pelleted and the eIF2 remaining in each supernatant fraction was resolved by SDS-PAGE and immunoblotted with two eIF2 subunit antibodies (inset, eIF2α and eIF2γ). Fraction bound at equilibrium was

*Figure 1 continued on next page*

*Figure 1 continued*

determined by quantification: total (lane 0 nM eIF2B) minus fraction remaining in the supernatant (graphs) and used calculate the dissociation constants (nM ± standard error (SE)) indicated. Assays were done either without nucleotide (C) or in the presence of either 1 mM GDP (D) or 1 mM GTP (E). (F). Cartoon of figure conclusion.

forms of eEF1A are >10 fold less stable ($K_d$ = 1.5 μM) (*Gromadski et al., 2007*). For the prokaryotic EF-Tu and EF-Ts, the nucleotide free complex is 1000-fold more stable ($K_d$ = 3 nM) than nucleotide bound forms ($K_d$ = 2–6 μM) (*Gromadski et al., 2002*). eIF2γ is structurally conserved with the tRNA-binding elongation factors eEF1A and EF-Tu as well as aIF2 found in archaea and all bind to both GDP/GTP and to tRNAs (*Schmitt et al., 2010*). However, unlike the translation elongation factors, eIF2/aIF2 each have three subunits (α-γ) and analyses indicate that both the eIF2α and β subunits also make important contacts with Met-tRNA$_i$ within the TC and modulate nucleotide binding to eIF2γ (*Huang et al., 1997*; *Naveau et al., 2013*; *Llácer et al., 2015*). eIF2αβ also influence interactions with eIF2B and eIF5, as indicated above.

The additional eIF2 and eIF2B subunit complexity over other translation G-proteins and GEFs suggested to us that eIF2B might not behave as a typical GEF. In addition a recent studies examining the EF-Tu interactions with its GEF, EF-Ts, suggest that EF-Ts can stimulate EF-Tu TC formation and can form an EF-Tu-GTP/tRNA/EF-Ts quaternary complex (*Burnett et al., 2013*, *2014*). These findings prompted us to investigate the interplay between eIF2B and eIF5 with eIF2-GTP and Met-tRNAi during eIF2 activation and its control by eIF2(αP). We report the unexpected finding that eIF2B binding affinity for eIF2 at steady state is not influenced by guanine nucleotides. We also report an antagonistic ability of eIF2B, where eIF2B can destabilise TC and compete with tRNAi for eIF2. We show that these new antagonistic roles for eIF2B are counteracted by eIF5, wherein eIF5 stabilises TC preventing the destabilizing effects of eIF2B. Hence eIF5 is required for robust rates of TC formation and transition initiation. Together these findings suggest that the competition between eIF2B and eIF5 for eIF2 continues even after TC formation and that TC/eIF5 complexes represent a robust product of nucleotide exchange. Finally we demonstrate that eIF2B can better compete with eIF5 for binding to TC formed with eIF2(αP), leading to increased TC destabilization. We propose these new interactions of eIF2B with TC reveal a second step of translation initiation regulated by eIF2(αP) and that these events ensure tight control of translation initiation.

## Results

### The affinity of eIF2B and eIF2 is not influenced by guanine nucleotides

GTP-binding protein interactions with their GEFs are typically modulated by the nucleotide bound to the G-protein, wherein the affinity for the nucleotide-free exchange intermediate is greatest and contributes to the forward direction of the G protein activation process (*Bos et al., 2007*). This preference has been found for translation elongation factors eEF1A and EF-Tu, that, like eIF2, also bind tRNAs during protein synthesis (*Gromadski et al., 2002*, *2007*). We investigated eIF2/eIF2B complex formation and in contrast to our expectations we found that, at steady state, eIF2 bound eIF2B with equal affinity independent of its nucleotide status. Our assay used Flag affinity resin to immobilize purified Flag tagged eIF2B over a wide range of concentrations (0–100 nM) incubated with limiting apo-eIF2 (2 nM eIF2 incubated to remove any co-purifying nucleotide, see Materials and methods) in the presence of 1 mM GDP, GTP or nucleotide free (*Figure 1B*). The fraction of eIF2 remaining unbound was quantified and used to calculate equilibrium dissociation constants ($K_d$s). $K_d$s showed minimal variations according to nucleotide status 26.6 nM (+GDP) to 32.2 nM (apo-eIF2) (*Figure 1C–E*). eIF2B has recently been shown to be a dimer (*Gordiyenko et al., 2014*; *Kashiwagi et al., 2016a*) and so capable of binding two eIF2 molecules per dimer. Our analyses provided no evidence for co-operative binding and the data fit well to a model where a 5-subunit eIF2B monomer binds one eIF2 heterotrimer (see Materials and methods). So in these and all following experiments our eIF2B concentrations assume a 5-subumit monomer forming a 1:1 complex with eIF2 and differ by two-fold from values that would be obtained by an eIF2B dimer. The measured eIF2/eIF2B $K_d$s are similar to the measured affinity between eIF5 and eIF2-GDP and

between eIF5 and TC (both ~23 nM) (*Algire et al., 2005*). Hence neither regulator of eIF2 shows a preference for eIF2 nucleotide status. Both of these eIF2 regulatory proteins bind via interactions with both eIF2$\beta$ and $\gamma$ (*Asano et al., 1999*; *Mohammad-Qureshi et al., 2007a*). As eIF5 has opposing GDI and GAP functions with different forms of eIF2, these observations open up the possibility that eIF2B has additional roles regulating eIF2-GTP.

## Competition between eIF2B and Met-tRNAi for eIF2-GTP

As eIF2B does not have enhanced affinity for nucleotide-free eIF2 (*Figure 1F*) or a reduced affinity for GTP bound eIF2, the GEF step alone cannot provide a driving force for TC formation to promote rounds of protein synthesis initiation. We hypothesized that eIF2B binding to eIF2-GTP might impact on the ability of eIF2 to form TC with Met-tRNAi and therefore promote eIF2-TC formation. Indeed recent studies with bacterial EF-Tu showed that its GEF, EF-Ts, could accelerate both the formation and decay of EF-Tu TC via a transient quaternary complex (*Burnett et al., 2014*). We used BOP-N-Met-tRNA$_i$ (tRNA probes) in a fluorescence spectroscopy assay where increasing concentrations of eIF2 and excess nucleotides enabled us to monitor TC formation in solution. Consistent with recent studies (*Jennings et al., 2016*), eIF2-GTP TC formed readily ($K_{d\ tRNAi}$=1.13 nM), while eIF2-GDP significantly reduced Met-tRNAi binding by ~50 fold ($K_{d\ tRNAi}$=55.5 nM; *Figure 2A*). Surprisingly, we found that the addition of eIF2B significantly inhibited TC formation in a concentration dependent manner (*Figure 2B*). Our assays contained 20 nM eIF2 and even a ten fold lower amount of eIF2B antagonised TC formation. Importantly, this assay was performed without GDP and with an excess of GTP making eIF2B GDP/GTP exchange redundant for TC formation. This experiment reveals that eIF2B competes with Met-tRNAi for binding to eIF2-GTP. The inhibitory effect of eIF2B on TC formation observed here is greater than predicted by the individual binding affinities of Met-tRNAi and eIF2B for eIF2-GTP and a standard competitive inhibition model (*Schön et al., 2011*). The reasons for this are not yet clear. One idea is that despite GTP presence in vast excess, eIF2B binding to eIF2 may promote GTP release, as in the absence of bound Met-tRNAi, the rate of spontaneous GTP release from eIF2 is high (*Figure 2—figure supplement 1*). Hence, as GTP loss impairs Met-tRNAi affinity significantly (*Figure 2A*), this could reduce the apparent affinity for Met-tRNAi in the presence of eIF2B.

In contrast to the large negative effect of eIF2B on TC formation, monitoring eIF2/eIF2B interactions in the presence of GTP and Met-tRNAi revealed that Met-tRNAi only minimally weakened the eIF2-GTP/eIF2B interaction ($K_{d\ eIF2}$=45.2 nM; *Figure 2C*) compared with eIF2-GTP/eIF2B affinity (28.3 nM; *Figure 1C*). Importantly Met-tRNAi remained in the supernatant in our binding assay (*Figure 2C*) and excess eIF2B did not alter BOP-N-Met-tRNAi fluorescence (*Figure 2—figure supplement 2*), indicating that eIF2B does not independently bind Met-tRNAi. To assess if eIF2B could disrupt pre-formed TC, we altered our assay set-up by pre-forming eIF2-TC to assess the ability of eIF2B to release BOP-N-Met-tRNAi. In the absence of added eIF2B, TC was stable during the course of our assays (0 nM eIF2B in *Figure 2D*), however the addition of eIF2B disrupted TC, altering tRNA fluorescence with an IC$_{50}$ = 53.4 nM (blue triangles, *Figure 2D*). Thus at steady-state, eIF2B impairs eIF2 TC formation rates and can disrupt pre-formed TC complexes. Hence eIF2B is a competitive inhibitor of TC formation. By analogy with EF-Ts (*Burnett et al., 2014*), one idea is that competition can be effected by formation of an unstable eIF2-TC/eIF2B intermediate (*Figure 2E*). These data suggest that eIF2 TC is not a final stable product of the eIF2B catalysed nucleotide exchange reaction and that an additional step is required to stabilize eIF2 TC.

## eIF5 stabilizes TC formation, preventing antagonism by eIF2B

In models of translation initiation eIF2 TC interacts with eIF5 and with eIF1, −1A and −3 to stimulate TC binding to 40 S ribosomes (*Hinnebusch, 2014*). Hence, unlike the translation elongation factor G protein TCs that bind directly to translating 80 S ribosome A-sites, additional protein factors modulate eIF2 TC downstream functions in translation initiation. We hypothesized that eIF5 may be a good candidate factor to modulate the apparent negative/competitive interfering role of eIF2B in destabilizing eIF2 TC (*Figure 2*) because eIF5 binds TC (*Algire et al., 2005*) to promote its interaction with eIF3 (*Singh et al., 2005*) and eIF1 and eIF4G (*Yamamoto et al., 2005*; *Luna et al., 2012*) and within the 48 S PIC eIF5 promotes start codon recognition during scanning and eIF2-GTP hydrolysis steps (*Huang et al., 1997*; *Luna et al., 2012*). Potentially at odds with this idea, we previously

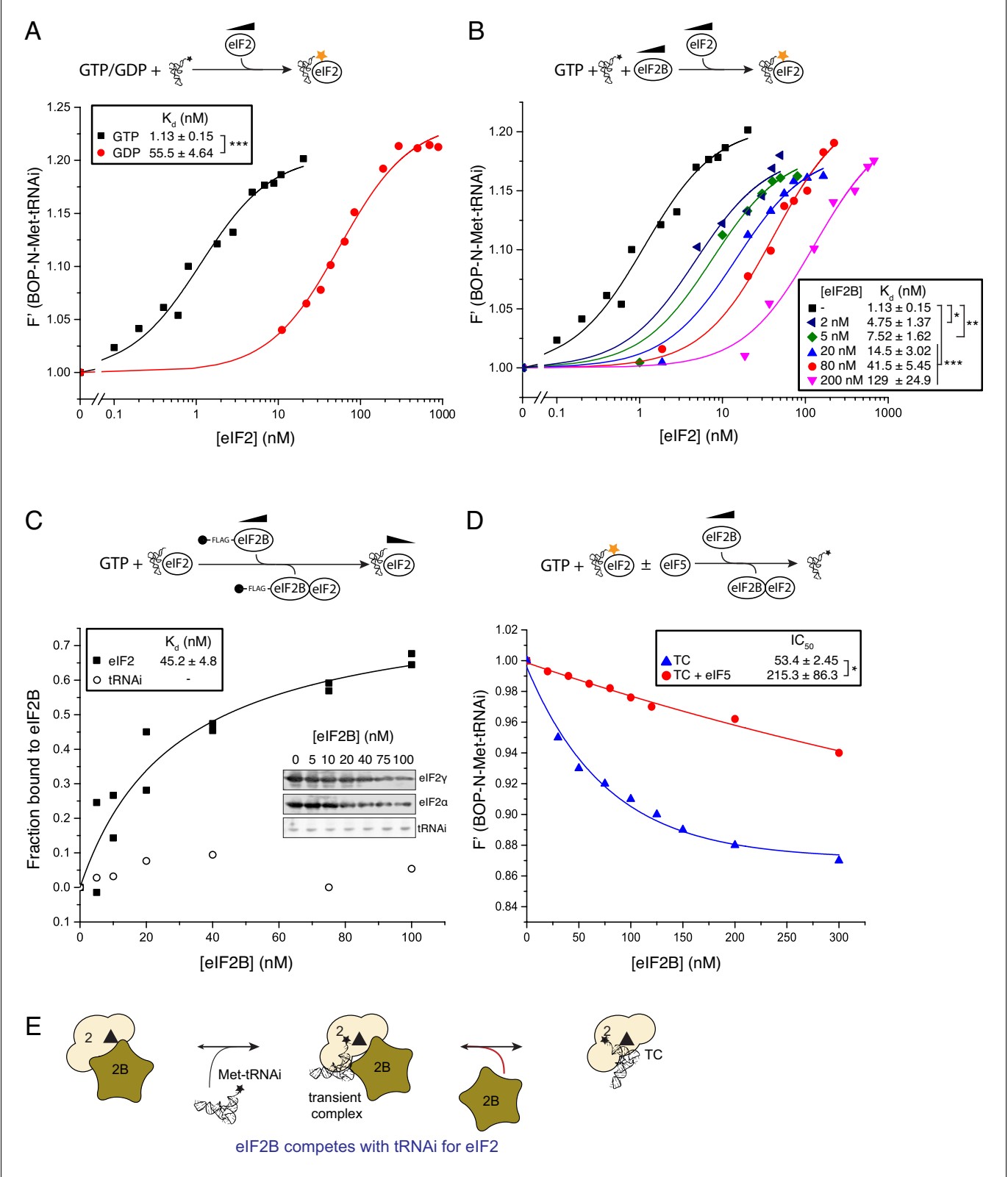

**Figure 2.** eIF2B competes with Met-tRNAi for binding to eIF2-GTP. (**A**) Binding curves titrating 20 nM BOP-N-Met-tRNAi with eIF2 in the presence of 1 mM GTP or GDP. Dissociation constants (nM) ± SE are indicated (inset). ***p<0.001 two-tailed T-test. (**B**). As in A only in the presence of 1 mM GTP ± increasing concentrations of eIF2B (0–200 nM). *p<0.05, **p<0.01, ***p<0.001 two-tailed T-test **C**). Supernatant depletion affinity capture as done in *Figure 1* with 1 mM GTP +2 nM Met-tRNAi with calculated affinity constants ± SE **D**). Dissociation curves for 20 nM BOP-N-Met-tRNAi pre-bound to

*Figure 2 continued*

eIF2 (20 nM) and saturating GTP (1 mM) upon titration of eIF2B (0–300 nM) (blue triangles). eIF5 (20 nM) was added to some reactions (red circles). Calculated $IC_{50}$ values ± SE for dissociation of TC are shown in inset box (nM). *p<0.05 two-tailed T-test. (E). Model for eIF2B competition with tRNA.

The following figure supplements are available for figure 2:

**Figure supplement 1.** Guanine nucleotide release from eIF2.

**Figure supplement 2.** eIF2B does not bind Met-tRNAi.

showed that eIF2B could readily displace eIF5 from eIF2-GDP binary complexes via its GDI-displacement factor (GDF) function ($IC_{50\ eIF2B}$=15.1 nM) (*Jennings et al., 2013*) (*Figure 1A*). How Met-tRNAi binding to eIF2/eIF5 would impact eIF2B GDF was not known.

We therefore examined the impact of eIF5 on the ability of eIF2B to destabilize eIF2-GTP and eIF2-TC formation. Pre-binding eIF2-GTP to eIF5, using 20 nM eIF5 (a 10-fold excess over eIF2 concentration in our assay) had only minimal impact on the eIF2-GTP/eIF2B interaction ($K_d$ = 37 nM, *Figure 3A*) compared to assays without eIF5 ($K_d$ = 28 nM; *Figure 1C*). This result is consistent with our previous report characterising the eIF2B GDF function that disrupts eIF2/eIF5 complexes (*Jennings et al., 2013*). In contrast, addition of eIF5 and Met-tRNAi significantly impaired the ability of eIF2B to bind eIF2-GTP ($K_d$ = 223 nM; *Figure 3B*). Similarly, in assays monitoring the binding of Met-tRNAi to eIF2-GTP, whereas eIF2B reduced the affinity for BOP-N-Met-tRNAi affinity for eIF2 ($K_d$ increased from 1.13 to 14.5 nM, *Figure 3C*), eIF2B and eIF5 together did not affect the Met-tRNAi affinity for eIF2 ($K_d$ = 1.43). Similarly, eIF5 alone had no impact ($K_d$ = 1.04). Finally, pre-incubation of TC with eIF5 effectively prevented TC destabilization by eIF2B ($IC_{50}$ = 215 nM; *Figure 2D*). Together these data indicate that eIF2B can disrupt eIF2-GTP/eIF5 and TC (eIF2-GTP/Met-tRNAi) but not the quaternary complex of (eIF2-GTP/Met-tRNAi/eIF5). They suggest a model for eIF2 recycling and TC formation that requires an additional step over the currently accepted pathway: the formation of TC/eIF5 complex that enables Stabilization of TC (STC), *Figure 3D*. These experiments define STC as a role for eIF5 to promote directionality to eIF2 recycling. By preventing pathway reversal by eIF2B, eIF5 STC function should help ensure eIF2 recycling proceeds in a forward direction.

## eIF5 STC function requires the eIF5-carboxyl terminal domain (CTD)

Next we wished to define the eIF5 domain requirements for stabilization of eIF2-TC. eIF5 is a single polypeptide with separate functional domains. Its amino terminal domain (NTD; residues 1–152) is required for GAP activity, while its carboxyl terminal domain (CTD; residues 241–405) is responsible for promoting interactions with eIF2 and other factors within the PIC (*Yamamoto et al., 2005*; *Luna et al., 2012*). The eIF5 NTD and CTD are joined by a linker region (LR; residues 153–240). The LR and CTD together are required for GDI activity (*Jennings and Pavitt, 2010a*). We assessed the ability of eIF5 domains to exhibit STC function in our TC formation assay. The eIF5-CTD alone was sufficient to prevent eIF2B destabilizing eIF2-TC equivalent to full-length eIF5 (*Figure 4A* and *Figure 4—source data 1A*, left panel). In contrast, the NTD alone was not able to stop eIF2B destabilizing eIF2-TC, as expected because the N-terminal domain has poor affinity for eIF2 (*Jennings and Pavitt, 2010a*). This result provides a clear distinction between STC function that requires only the eIF5 CTD and the previously described eIF5 GDI function that requires the LR in addition to the CTD.

## eIF2Bε alone destabilizes TC, even in the presence of eIF5

As indicated in the introduction, eIF2B is a large multi-subunit complex encoded by five different genes. We wished to determine which was responsible for destabilization of eIF2 TC. eIF2B GEF activity requires eIF2Bε and is boosted by complex formation with eIF2Bγ. Similarly both subunits are needed for GDF (*Jennings et al., 2013*). We found that both eIF2Bε alone and eIF2Bγε subcomplexes were as or more effective than full eIF2B complexes in competing with BOP-N-Met-tRNAi binding to eIF2-GTP (*Figure 4B* and *Figure 4—source data 1B*). For example 20 nM eIF2B

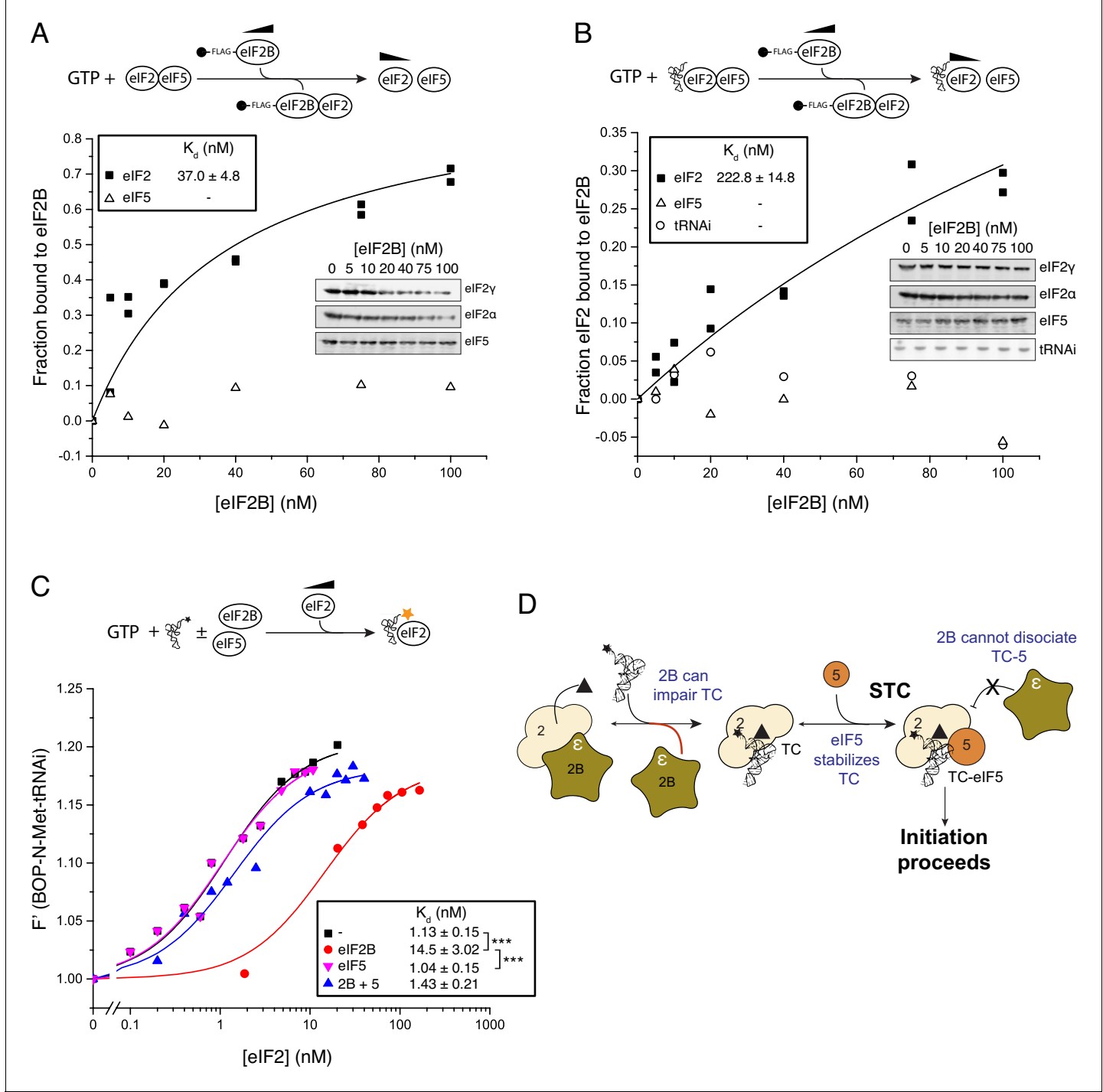

**Figure 3.** eIF5 stabilizes Met-tRNAi binding to TC. (**A**) eIF2/eIF2B equilibrium binding assay in the presence of eIF5 as described in legend to *Figure 1*. eIF2 (2 nM) was pre-bound (prior to mixing with eIF2B) with eIF5 (20 nM) and GTP (1 mM) then mixed with increasing concentrations of eIF2B immobilised on anti-Flag resin. (**B**). As in A, but with Met-tRNAi (20 nM) also added prior to eIF2B. (**C**). Binding curves titrating BOP-N-Met-tRNAi (20 nM) with eIF2 in the presence of GTP (1 mM) as in *Figure 2A*, but with eIF5 (20 nM), eIF2B (20 nM) or both eIF5 and eIF2B (20 nM each). Dissociation constants (nM) are indicated. ***p<0.001, two-tailed T-test. (**D**). Model for eIF5 stabilization of TC.

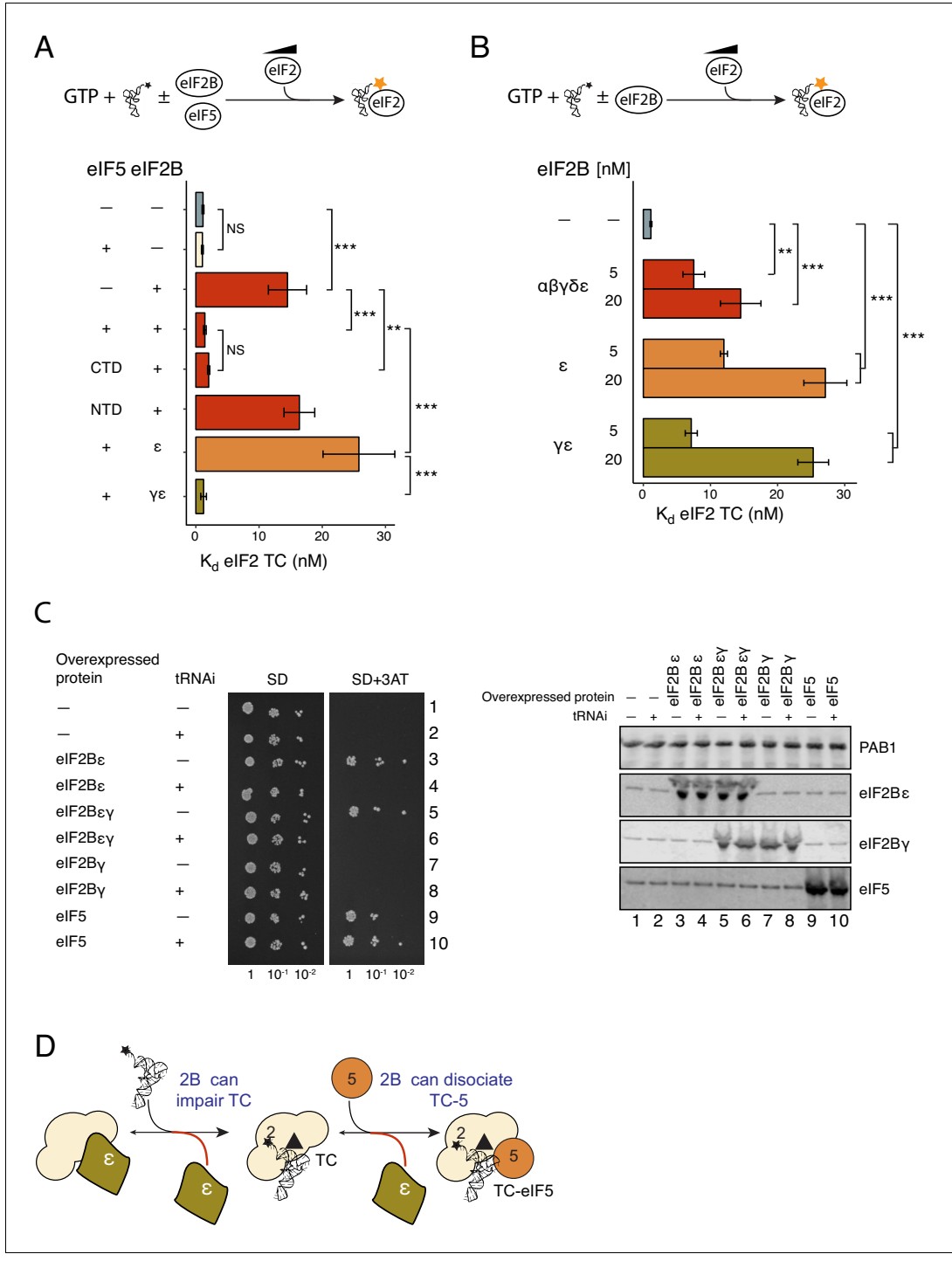

**Figure 4.** eIF2Bε antagonises eIF5 STC function. (**A**) $K_d$ measurements from eIF2-TC formation assays shown in *Figure 4—source data 1*. Experiments were performed as in *Figure 2A*.±20 nM eIF2B ±20 nM eIF5, eIF5-NTD or eIF5-CTD. (**B**). As panel A except ±5 nM or 20 nM of full eIF2B complex, eIF2Bγε subcomplexes or eIF2B epsilon alone. (**C**). Left, Serial dilution growth assay of *gcn2Δ* yeast cells bearing multi-copy plasmids overexpressing the indicated combination of tRNAi (*IMT4*) and eIF2Bε (*GCD6*), eIF2Bγ (*GCD1*), eIF5 (*TIF5*) or eIF2Bεγ (*GCD6 + GCD1*) grown on minimal and 3AT medium. Right, Western blot of strains used confirming overexpression of indicated proteins. (**D**). Model showing eIF2Bε antagonism of TC/eIF5 and TC. **p<0.01, ***p<0.001, NS non-significant (p≥0.05), two-tailed T-test.

The following source data is available for figure 4:

*Figure 4 continued on next page*

*Figure 4 continued*

**Source data 1.** Destabilisation of TC requires eIF2Bε and stabilisation of TC requires eIF5-CTD.

complexes reduced Met-tRNAi affinity by ~10 fold from 1.13 to 14.5 nM, while 20 nM eIF2Bε alone reduced Met-tRNAi affinity by ~20 fold (to 27 nM; *Figure 4B*). To further explore eIF5 and eIF2B antagonism we repeated these TC formation assays with 20 nM eIF2B subcomplexes and 20 nM eIF5. Here, eIF2Bε alone and eIF2Bγε subcomplexes behaved very differently to the presence of eIF5. eIF2Bγε subcomplexes were out-competed by eIF5 STC function and behaved like full eIF2B complexes. In contrast eIF2Bε was unaffected by eIF5 and was able to impair TC formation (*Figure 4A* and *Figure 4—source data 1A*, right panel). This result suggests that an unexpected and novel function of eIF2Bγ is to prevent eIF2Bε antagonising TC/eIF5 complexes and thereby facilitate eIF5 STC activity.

It was shown previously that overexpression of eIF2Bε in yeast cells caused an unexpected phenotype. Excess eIF2Bε gene dosage promotes Gcn2-independent activation of the general amino acid control response (GAAC), the yeast counterpart to the mammalian ISR (*Richardson et al., 2004*). Activation of GAAC under these conditions implies cells have reduced active eIF2-TC levels that in turn stimulate *GCN4* translation to levels that can overcome an imposed amino acid limitation (*Hinnebusch, 2005*; *Dever et al., 2016*). This result is not expected from eIF2Bε's known functions in eIF2 recycling, but is consistent with the results described above and shown in *Figure 4A*. Our data predict that by increasing Met-tRNAi gene dosage we might suppress the aberrant GAAC response by mass action. We therefore tested this idea, monitoring the growth of *gcn2Δ* cells on 3-aminotriazole containing medium (SD + 3AT) to assess GAAC activation. In accord with our prediction, increasing Met-tRNAi gene dosage suppressed the aberrant Gcn2-independent GAAC response associated with excess eIF2Bε gene dosage (*Figure 4C*, Compare rows 3 and 4). We also found that co-overexpressing eIF2Bγ with ε resulted in the same Gcn2-independent growth and suppression by excess Met-tRNAi (rows 5 and 6). As expected expressing eIF2Bγ alone did not confer the GAAC phenotype, (*Figure 4C*, rows 7 and 8), while excess eIF5 did (row 9). Excess eIF5 was previously shown to antagonise eIF2Bε (*Singh et al., 2006*) and in line with expectations this phenotype was not suppressed by excess Met-tRNAi gene dosage (row 10). Western Blotting confirmed that analysis excess Met-tRNAi did not alter protein overexpression levels (*Figure 4C*, right panels) and quantification indicated that eIF2Bε was overexpressed ~24 fold over wild-type, while eIF2Bγ was overexpressed ~12 fold over wild-type levels. Hence the cells shown in rows 5 and 6 (*Figure 4C*) likely contain a mix of excess eIF2Bγε complexes and excess free eIF2Bε that both contribute to the observed phenotype. Together with the biochemical analyses, these data show that eIF2Bε alone can act as a rogue factor to antagonise TC and TC/eIF5 complex formation (*Figure 4D*). These results are consistent with the idea that eIF2B subunit complexity contributes to effective eIF2 recycling by reducing the ability of eIF2Bε to antagonise the recruitment of eIF2 TC by eIF5 and hence promote translation initiation.

## eIF2B can disrupt eIF2(αP) TC/eIF5 complexes

Because phosphorylation of eIF2α at serine 51 by eIF2α kinases is a universal and potent inhibitor of protein synthesis, we investigated whether this would have any impact on these new activities of eIF2B and eIF5. As phosphorylated eIF2 [eIF2(αP)] can form a tight complex with eIF2B via binding to its regulatory αβδ subunits (*Pavitt et al., 1998*; *Krishnamoorthy et al., 2001*), we hypothesized that this may enable eIF2B to further antagonise eIF5/TC complexes. We used purified PKR to phosphorylate eIF2 and employed Phos-tag SDS-PAGE gel immunoblots to demonstrate that eIF2α was phosphorylated to at least 80% (*Figure 5—figure supplement 1A*). To assess the impact of phosphorylated α on eIF2 kinetic parameters we repeated a large selection of our established assays with this new substrate. Importantly, eIF2(αP) did not alter the intrinsic $K_{off\ GDP}$ observed in our assays, but did prevent eIF2B stimulation of BODIPY-GDP release in line with previous findings (*Figure 5—figure supplement 1B*). In addition, eIF2(αP) exhibited a 8–10-fold enhanced affinity for eIF2B over non-phosphorylated eIF2, irrespective of eIF2 nucleotide status (*Figure 5A* and *Figure 5—source data 1* panels A-C). Phosphorylation did not impact the affinity of eIF2 for BOP-N-Met-tRNAi, either

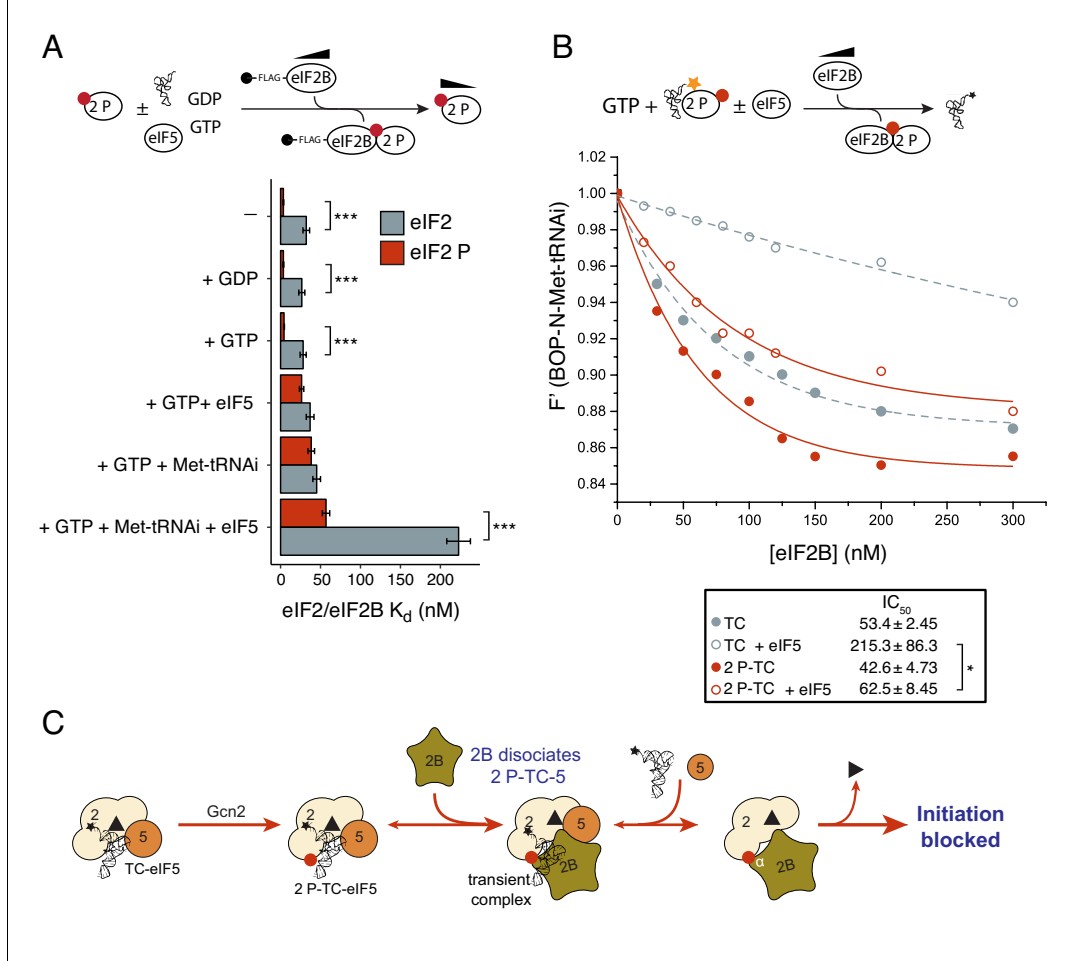

**Figure 5.** eIF2B antagonizes eIF2(αP)-TC/eIF5 complexes. (**A**) $K_d$ measurements ± SE from eIF2(αP)/eIF2B complex formation assays shown in *Figure 5—source data 1* and compared with measurements made with non-phosphorylated eIF2 shown in previous figures. (**B**). Dissociation curves for 20 nM BOP-N-Met-tRNAi pre-bound to eIF2(αP) (20 nM) and saturating GTP (1 mM) upon titration of eIF2B (0–300 nM) (red circles). eIF5 (20 nM) was added to some reactions (open circles). Data shown in *Figure 2D* is reproduced for comparison in gray symbols. Calculated $IC_{50}$ values ± SE are shown in the box. (**C**). Model for eIF2B inhibition of eIF2(αP)-TC/eIF5 complexes. *p<0.05, ***p<0.001, two-tailed T-test.

The following source data and figure supplements are available for figure 5:

**Source data 1.** Binding between eIF2(αP) and eIF2B in the presence of different ligands.
**Figure supplement 1.** Phosphorylation of eIF2α and its impact on eIF2B activity.
**Figure supplement 2.** eIF2(αP) affinity for Met-tRNAi.

in the presence of GTP ($K_{d\ tRNA}$1.35 nM) or GDP ($K_{d\ tRNA}$49.3 nM) (*Figure 5—figure supplement 2*). The lack of impact of phosphorylation on Met-tRNAi affinity eIF2 is in line with expectations given that structural analysis of eIF2-tRNA interactions indicates that there is no direct contact between the serine 51 of eIF2α and Met-tRNAi (*Llácer et al., 2015*) and because in yeast mutations in eIF2 or eIF2B subunits permit cells to grow at normal rates with very high eIF2αP levels (*Pavitt et al., 1997*; *Vazquez de Aldana and Hinnebusch, 1994*; *Vazquez de Aldana et al., 1993*): experiments imply-ing that eIF2αP inhibits eIF2B function only.

To examine completion between eIF2B, eIF5 and Met-tRNAi for eIF2(αP) we repeated our previ-ous interaction assays. eIF2B had only a modest 1.2–1.4-fold enhanced affinity for eIF2(αP)-GTP over eIF2-GTP in the presence of excess Met-tRNAi or eIF5 alone (*Figure 5A* and *Figure 5—source data*

*1*, panels D-E). In contrast when 10-fold molar excess of both Met-tRNAi and eIF5 over eIF2(αP) were both included in the binding assay, eIF2(αP)/eIF2B complexes formed with 4-fold enhanced affinity over non-phosphorylated eIF2 ($K_{d\ eIF2}$ =57 nM vs 223 nM for unphosphorylated eIF2/eIF2B (*Figure 5A* and *Figure 5—source data 1*, panel F). These experiments suggest that eIF2B is better equipped to disrupt TC/eIF5 complexes when eIF2 is phosphorylated, than when non-phosphorylated. To quantify this we preformed TC with eIF2(αP) ± eIF5 and asked if eIF2B could disrupt Met-tRNAi binding to eIF2. As shown in *Figure 5B*, eIF2B is able to displace Met-tRNAi from eIF2(αP)-TC/eIF5 complexes (red open circles) ~4x as well as it can with non-phosphorylated TC / eIF5 (gray open circles). Indeed Met-tRNAi displacement from eIF2(αP)-TC/eIF5 complexes by eIF2B proceeded as effectively as Met-tRNAi displacement from TC complexes lacking eIF5 (gray filled circles in *Figure 5B*). Together these data are consistent with the idea that in addition to regulation of eIF2B GEF activity, eIF2B can also further inhibit translation initiation by disrupting eIF2(αP)-TC/eIF5 complexes and forming eIF2(αP)-GTP/eIF2B complexes (*Figure 5C*). Unlike bound GDP, GTP readily dissociates from eIF2(αP) (*Figure 5—figure supplement 1B*) which should further impede Met-tRNAi re-acquisition. We propose that these inhibitory interactions with eIF2(αP)-GTP act as secondary 'fail-safe' mechanisms to ensure tight-regulation of translation by phosphorylation of eIF2.

## Discussion

### eIF2B does not preferentially bind nucleotide free eIF2

We have evaluated interactions between eIF2B and eIF2, Met-tRNAi and eIF5 in the formation and regulation of eIF2-GTP/Met-tRNAi ternary complexes. Our studies reveal five key novel findings about the mechanisms driving translation initiation and its regulation by eIF2αP, a key element of the ISR. Firstly we show that eIF2B affinity for eIF2 is not governed by the nucleotide bound (*Figure 1*). This is highly unusual as GEFs typically bind their G proteins with higher affinity for a nucleotide-free form. Such a binding mode helps drive the extraction of the inactive GDP and promote binding of GTP (*Bos et al., 2007*). This lack of specificity of eIF2B for nucleotide-free eIF2 is likely because eIF2 has additional subunits over other translational G proteins. It is known that eIF2*β* is important for interacting with both regulatory factors (*Asano et al., 1999*), hence eIF2 regulator interactions are not dependent solely on the nucleotide-binding eIF2*γ* subunit. A likely consequence of a lack of preference for the nucleotide free form is that eIF2B-promoted GDP release and subsequent binding of GTP does not by itself provide a strong forward momentum to promote eIF2B release from eIF2-GTP and drive TC formation.

### eIF2B competes with Met-tRNAi for eIF2-GTP

When investigating the influence of eIF2B on eIF2-TC formation we found that eIF2B and Met-tRNAi compete for eIF2-GTP. eIF2B therefore paradoxically promotes translational activation via GEF action and also acts as a competitive inhibitor impeding TC formation. We assume that because eIF2B concentrations in vivo are lower than eIF2 and Met-tRNAi (*Singh et al., 2007*; *von der Haar and McCarthy, 2002*), that tRNAi abundance helps favour the forward reaction during times of optimal growth. The TC destabilizing activity of eIF2B appears important when translational down-regulation by eIF2 phosphorylation is required, (see later discussion). Hence the negative impact of eIF2B on TC formation is likely the price cells pay to permit tight regulation of TC levels by eIF2B. eIF2B and Met-tRNAi competition for eIF2 only requires eIF2Bε (*Figure 4B*). This is consistent with the idea that Met-tRNAi and eIF2Bε compete for an overlapping or shared binding site on eIF2 and that binding is mutually exclusive. Such competition between eIF2B and Met-tRNAi helps to explain why excess eIF2Bε aberrantly activates *GCN4* translation. Although excess eIF2Bε enhances eIF2B GEF activity (*Pavitt et al., 1998*), this is not reflected by *GCN4* activity in vivo which is elevated in cells lacking gcn2 but with excess eIF2Bε. Gcn2-independent growth on 3AT medium is termed a Gcd⁻ phenotype, which signals limiting active TC levels. Genetic suppression of the Gcd- phenotype by excess tRNAi gene dosage (*Figure 4C*) is in accord with a model where eIF2Bε and Met-tRNAi can compete for eIF2-GTP.

## eIF5 stabilizes TC

eIF5 is known to promote eIF2-TC recruitment to other initiation factors (*Asano et al., 2000*; *Yamamoto et al., 2005*; *Luna et al., 2012*). Here we found that eIF5 stabilizes eIF2-TC, effectively preventing eIF2B releasing eIF2 from TC/eIF5 complexes (*Figure 3*). These data are consistent with free eIF5 providing a driving force stabilizing TC in preparation for translation initiation and preventing eIF2B antagonizing eIF2-TC. One interpretation of this data is that eIF2-TC/eIF5 complexes represent the final product of nucleotide exchange rather than TC itself. A further implication of our findings is that by down-regulating eIF5 levels or activity it may be possible to influence the availability of TC/eIF5 complexes for translation. eIF5 levels are regulated through uORFs in mammalian cells (*Loughran et al., 2012*), and eIF5 mimic proteins have been described that can antagonise eIF2/eIF5 functions, suggesting additional regulatory inputs could potentially control TC activity (*Singh et al., 2011*; *Kozel et al., 2016*). Only the eIF5-CTD was required to stabilize TC (*Figure 4A*). This is consistent with prior results that eIF5-CTD is critical for assembling the eIF2 multifactor complex and 43S PIC (*Asano et al., 2000*; *Yamamoto et al., 2005*) and shows that TC stabilization does not require the eIF5 linker-region, which is necessary for eIF5 GDI function with eIF2-GDP (*Jennings and Pavitt, 2010b*).

## eIF2Bγ plays critical roles to promote directionality to eIF2 recycling

We found recently that the eIF2Bγε subcomplex is necessary for eIF2B to actively dissociate eIF2 from eIF2/eIF5 complexes (*Jennings et al., 2013*). Neither eIF2Bε or eIF2Bγ alone could stimulate eIF5 release. eIF2Bγε sub-complexes also have enhanced GEF activity compared with eIF2Bε alone, activity equivalent to eIF2B full complexes (*Pavitt et al., 1998*; *Jennings et al., 2013*). Here we show that eIF2Bγ has a third important role, as it prevents eIF2Bε destabilizing TC/eIF5 complexes (*Figure 4A*). We interpret these observations as indicating that eIF2Bγ plays multiple critical roles ensuring efficient eIF2 recycling. Firstly it facilitates release of eIF2-GDP from eIF2-GDP/eIF5 complexes (GDF), next it boosts GDP release (GEF) and finally it prevents eIF2Bε destabilizing TC/eIF5 complexes. Together these data show eIF2Bγ is critical for efficient eIF2 recycling through multiple steps under conditions of active growth.

## eIF2(αP)-GTP complexes can be regulated by eIF2B

eIF2(αP) inhibits eIF2B GEF activity (*Pavitt et al., 1998*; *Jennings et al., 2013*) by forming a tight complex between eIF2(αP)-GDP and eIF2B via binding eIF2α to its regulatory αβδ subunits (*Pavitt et al., 1998*; *Krishnamoorthy et al., 2001*; *Kashiwagi et al., 2016b*). However our experiments provide evidence that eIF2B can also disrupt eIF2(αP)-GTP functions independently of blocking eIF2B nucleotide exchange. eIF2B can destabilize eIF2(αP)-TC/eIF5 complexes releasing both eIF5 (*Figure 5—source data 1*, panel F) and Met-tRNAi (*Figure 5B*). We interpret these findings as evidence of a secondary control system, or 'failsafe' mechanism to enable free eIF2B to mop up any eIF2(αP) in its vicinity, even when the latter was bound to Met-tRNAi and eIF5. As shown in cartoon form in *Figure 6*, these activities likely allow eIF2B to wind-back initiation complex intermediates to boost eIF2(αP)/eIF2B complexes. Unlike its tight interaction with GDP, GTP-bound eIF2, is very unstable (*Figure 2—figure supplement 1*). Release of GTP from inhibited complexes would further ensure tight control of translation. We envisage this mechanism as providing additional safeguard to rapidly limit protein synthesis upon the onset of eIF2(αP), rather than restricting protein synthesis inhibition to eIF2(αP)-GDP complexes.

How effective this additional mechanism of control is in vivo is not easy to evaluate, as there is no tool currently available to decouple regulation of GTP complexes from GDP complexes. eIF2B is generally thought present in limiting concentrations compared with eIF2, eIF5 and Met-tRNAi, and this is consistent both with maintaining active translation and limiting the ability of eIF2B to compete with Met-tRNAi when optimal growth and translation are required. Although it has been observed that eIF2B can be concentrated within cellular granules or bodies whose role is not clear, but which likely alter local relative eIF2:eIF2B ratios. For example eIF2B is found in diffusible cytoplasmic foci termed eIF2B bodies or filaments (*Campbell et al., 2005*; *Taylor et al., 2010*; *Noree et al., 2010*), the abundance of which has been shown to be regulated by some cell stresses (*Petrovska et al., 2014*). The eIF2 kinase Perk is an ER-membrane tethered kinase thought to preferentially regulate the translation of ER-localized protein synthesis during the unfolded protein response. In addition

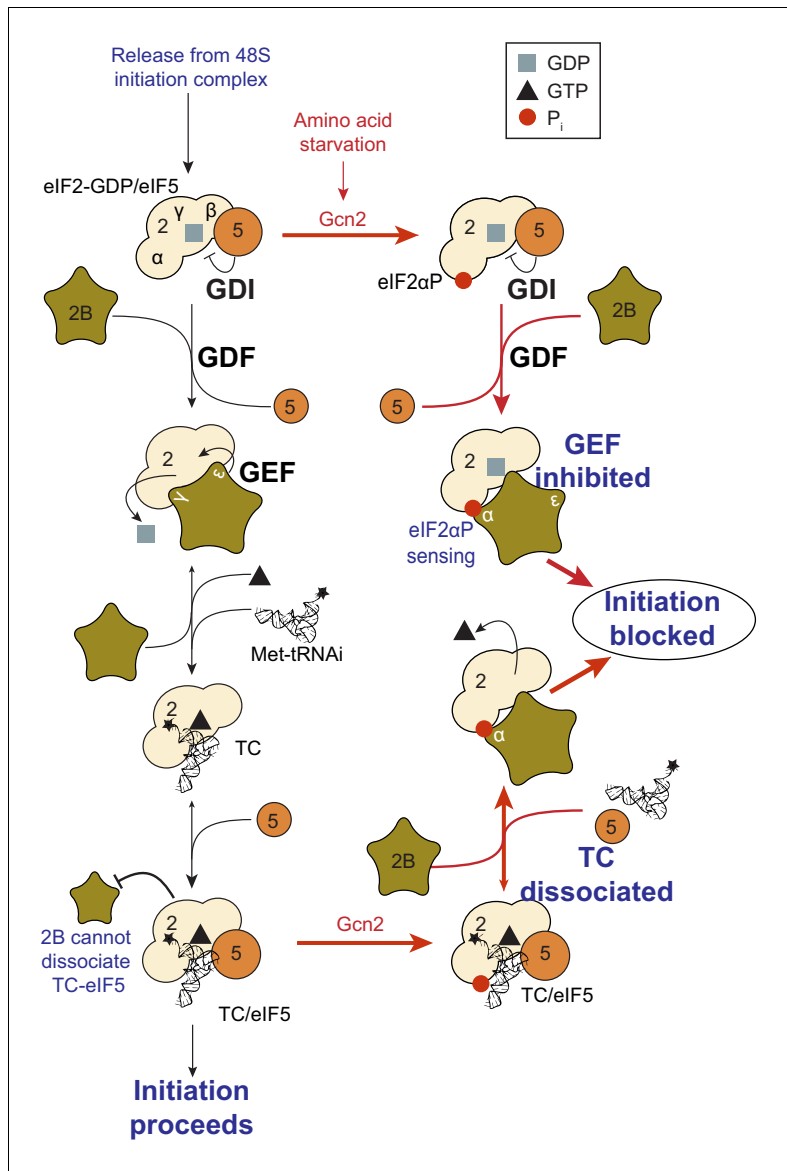

**Figure 6.** Model summary of new activities of eIF2 and eIF5. A summary model of interactions between eIF2, eIF2B, eIF5 and Met-tRNAi to generate eIF2-TC/eIF5 complexes and their inhibition by eIF2(αP). See text for details.

stress granules form upon a wide variety of cellular stresses in a variety of cell types including stresses that generate phosphorylated eIF2 (*Buchan and Parker, 2009*; *Kedersha et al., 2013*). The ratios of translation factors in these various granules likely differ from their overall relative abundances in the cytoplasm. Such variations may permit lower affinity interactions to be biologically meaningful. Together, these findings provide a functional framework to our understanding of eIF2B complexity and its multiple roles in eIF2 recycling and suggest additional eIF2(αP) controlled steps operate within the translation initiation pathway.

## Materials and methods

### Yeast genetics

Strain GP7124 (*MATα ura3-52 leu2-3 leu2-112 ino1 gcn2Δ sui3Δ trp1Δ::hisG ura3-52::P$_{HIS4}$-LacZ* pAV2443 [Flag-*SUI3 TRP1* CEN]) was grown in yeast extract peptone dextrose medium as described (*Amberg et al., 2005*) and transformed using lithium acetate method (*Gietz and Woods, 2006*) with plasmids pAV2330 [*IMT4 URA3* 2 μm], pAV1754 [*GCD6 LEU2* 2 μm], pAV1413 [*GCD6 GCD1 LEU2* 2 μm], pAV1875 [*TIF5*-FLAG *LEU2* 2 μm], pAV1162 [*GCD1 LEU2* 2 μm] or empty vector plasmid controls. Transformed strains were 10-fold serial diluted and spotted onto synthetic dextrose (SD) medium ±10 mM 1,2,3-aminotriazole and grown at 30°C.

### Cell-extracts, SDS-PAGE and immuno-blotting

For protein expression analysis, strains were grown in selective media to an $A_{600}$ of 0.6 then harvested by centrifugation at 5000 x *g*. Cells were resuspended in lysis buffer (30 mM HEPES pH 7.4, 100 mM KCl, 0.1 mM MgCl$_2$, 10% glycerol + EDTA-free protease inhibitors (Thermo Fisher Scientific, Loughborough, United Kingdom). 200 μl of acid washed glass beads (Sigma-Aldrich, Poole, United Kingdom were added and cells were lysed by using a FastPrep (MP Bio, Santa Ana, CA) for 3 × 20 s at 6.5 ms$^{-1}$ with cooling on iced water for 5 min between cycles. Cell debris was removed from the cell extract by centrifugation at 10,000 × *g* for 15 min at 4°C. SDS-PAGE and immuno-blotting were performed as previously described (*Jennings and Pavitt, 2010a*) using specific antibodies for eIF2α, eIF2γ, eIF2Bε, eIF5, eIF2Bγ, Flag-M2 (Sigma-Aldrich, Poole, United Kingdom), PAB1 (Encor Biotechnology, Gainesville, FL), ser-51 phosphorylated eIF2α (Abcam, Cambridge, UK), and eIF5. Secondary antibody probing and quantification was performed using IRDye 800CW goat anti-rabbit IgG with an Odyssey Fc imaging system (Li-Cor, Cambridge, United Kingdom).

### Protein purification

eIF2 was purified from yeast strain GP3511 as previously described (*Pavitt et al., 1998*). To obtain apo-eIF2 free from nucleotide, eIF2 was dialysed with EDTA (30 mM HEPES, 100 mM KCl, 1 mM DTT, 1 mM EDTA, 10% glycerol, pH 7.4) then with magnesium (30 mM HEPES, 100 mM KCl, 1 mM DTT, 0.1 mM MgCl$_2$, 10% glycerol, pH 7.4). GST-eIF5 was purified from *Escherichia coli* as described (*Jennings and Pavitt, 2010a*). Flag-eIF2B complexes, subunits and sub-complexes were purified from yeast, as described (*Mohammad-Qureshi et al., 2007b*) and (*Jennings et al., 2013*). Flag-PKR was also purified from yeast as previously described (*Jennings et al., 2013*)

### eIF2 phosphorylation

Purified eIF2 was phosphorylated using purified PKR. Typically, 5 μg of eIF2 was incubated with 0.3 μL PKR, 0.1 mM ATP, and 5 mM NaF for 15 min at room temperature. To assess the level of phosphorylation, samples were resolved by SuperSep Phos-tag (Wako, Osako, Japan) SDS-PAGE to separate phosphorylated and nonphosphoylated eIF2α prior to immunoblotting with eIF2α-specific antisera.

### Steady state fluorescence

To monitor Met–tRNA$_i$ binding to eIF2, 20 nM BOP-N-Met-tRNA$_i$ (tRNA Probes, College Station, TX) in 180 μl of assay buffer (30 mM HEPES, 100 mM KCl, 10 mM MgCl2, pH7.4) was measuring using a Fluoromax-4 spectrophotometer (Horiba, Stanmore, United Kingdom) (490 nm excitation, 509 nm emission). Change in fluorescence intensity was measured upon addition of increasing amounts of apo-eIF2, incubating for 5 min at room temperature each time. Each measurement was blanked against a control without nucleotide to account for any affect of eIF2 and data were corrected for dilution effects caused by volume addition and normalised to starting values before being fitted to a single site binding model: $y = 1 + [(\Delta F_{max} - 1)*(x/(x + K_d))]$ to obtain the dissociation constant ($K_d \pm$ SE). To monitor dissociation of TC, 20 nM BOP-N-Met-tRNA$_i$ was premixed with eIF2 +/- eIF5 in 180 μl of assay buffer before monitoring fluorescence intensity. Change in intensity was then monitored upon addition of increasing amounts of eIF2B, incubating for 5 min at room temperature each time. Each measurement was blanked and volume corrected. Data from 7–13 individual experiments was fitted to an exponential curve to calculate IC$_{50}$ values ± SE.

## eIF2-eIF2B equilibrium binding assay

To monitor the interaction between eIF2 and eIF2B, increasing amounts of eIF2B was pre-bound to 50 µl of anti-Flag M2 resin (Sigma-Aldrich, Irvine, United Kingdom). 2 nM of apo eIF2 was added to the Flag resin (±1 mM nucleotides, 20 nM eIF5, 2 nM met-tRNAi) in a total volume of 1 ml and bound for 1 hr at 4°C. Flag resin was pelleted and the eIF2 remaining in each supernatant fraction was concentrated (SpeedVac concentrator, Thermo Fisher Scientific, Loughborough, United Kingdom) then resolved by SDS-PAGE and immunoblotted with two eIF2 subunit antibodies (inset, eIF2$\alpha$ and eIF2$\gamma$). Fraction bound at equilibrium was determined by quantification and then subtracting the amount remaining upon addition of eIF2B from the total using a control where no eIF2B was added. Data from seven individual experiments were fitted to $[(A*x)/(K_d+x)]$ to calculate the equilibrium dissociation constant ($K_d$) $\pm$ SE. eIF5 in the supernatant was also probed using a specific antibody. Met-tRNAi in the supernatant was monitored by semi-quantitative RT-PCR using the oligonucleotide primers IMTF (AGCGCCGTGGCGCAGTGGAAGCGCGCA) and IMTR (TAGCGCCGCTCGGTTTCGA TCCGAG), Onestep RT-PCR Kit (Qiagen Ltd, Manchester, United Kingdom).

## GDP dissociation assay

Fluorescent eIF2•BODIPY-GDP binary complex was formed by incubating apo-eIF2 with a 2x excess of BODIPY-FL-GDP (Thermo Fisher Scientific, Loughborough, United Kingdom) for 20 min at room temperature. Excess nucleotide was removed by passing through a G-50 Sephadex column (GE Healthcare, Little Chalfont, United Kingdom). Labelling efficiency was calculated to exceed 90%. To measure GDP release, 20 nM eIF2•BODIPY-GDP was quickly mixed with 1 mM of unlabelled GDP ($\pm$ eIF2B and $\pm$ GST-eIF5) in 180 µl of assay buffer (30 mM HEPES, 100 mM KCl, 10 mM MgCl$_2$, pH 7.4) and fluorescence intensity was continuously measured using a Fluoromax-4 spectrophotometer (Horiba, Stanmore, United Kingdom) (490 nm excitation, 509 nm emission, 0.1 s integration time). 5 nM of eIF2B was added to stimulate nucleotide exchange. Experimental data were fitted to exponential dissociation curves to determine the rate constants ($K_{off}$) $\pm$ SE.

## Statistics

To determine statistical significance, standard errors (SE) reported from nonlinear curve regression were compared. Degrees of freedom were calculated as the sum of the data points in each fit minus the sum of the variables fit (two variables per fit, totaling four in all cases). T Scores were calculated as $T=(Fit1-Fit2)/\sqrt{(SE^1+SE^2)}$. $P$ values were then calculated based on a two-tailed T-test.

# Additional information

### Funding

| Funder | Grant reference number | Author |
| --- | --- | --- |
| Biotechnology and Biological Sciences Research Council | BB/L020157/1 | Graham D Pavitt |
| Biotechnology and Biological Sciences Research Council | BB/M006565/1 | Graham D Pavitt |
| Biotechnology and Biological Sciences Research Council | BB/L000652/1 | Graham D Pavitt |

The funders had no role in study design, data collection and interpretation, or the decision to submit the work for publication.

### Author contributions

MDJ, Conceptualization, Formal analysis, Investigation, Methodology, Writing—original draft, Writing—review and editing; CJK, Investigation, Writing—review and editing; TA, Investigation, Methodology, Writing—review and editing; GDP, Conceptualization, Resources, Formal analysis, Supervision, Funding acquisition, Investigation, Methodology, Writing—original draft, Project administration, Writing—review and editing

## Author ORCIDs

Graham D Pavitt, http://orcid.org/0000-0002-8593-2418

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
