## [Decision Letter]

Thank you for submitting your article "Fail-safe control of translation initiation by dissociation of eIF2α phosphorylated ternary complexes" for consideration by *eLife*. Your article has been reviewed by two peer reviewers, and the evaluation has been overseen by a Reviewing Editor and James Manley as the Senior Editor. The following individual involved in review of your submission has agreed to reveal his identity: Leos Shivaya Valasek (Reviewer #2).

The reviewers have discussed the reviews with one another and the Reviewing Editor has drafted this decision to help you prepare a revised submission.

Summary:

This manuscript addresses the mechanisms by which phosphorylation of eIF2 can control key mechanistic steps in translational control. The eIF2 combines with GTP and Met-tRNA(iMet) and is critical for ribosome selection of initiation codons. In the Integrated stress response (ISR), phosphorylation of eIF2 (α subunit) in response to cellular stress conditions sharply reduces the ability of its exchange factor (eIF2B) to recycle eIF2-GDP to GTP, hence lowering translation initiation. This manuscript addresses two related questions that are central to the ISR. First, can phosphorylated eIF2 function in the ternary complex during translation initiation? Second, if phosphorylated eIF2 can function in translation initiation processes, can it regulate key features including selection of the context of initiation codon. The later question has been proposed to explain phosphorylated eIF2 induction of preferential translation of ISR genes via a single regulatory uORF. This manuscript argues persuasively that the answer to both questions is no. Importantly, the manuscript shows that in addition to the eIF2B exchange function that is subject to regulation by phosphorylated eIF2, eIF2B can bind with phosphorylated eIF2, restricting it from associating with eIF5 (GAP and GDI) and as a consequence destabilizing the eIF2 ternary complex. Hence the fail-safe control in the ISR that is mentioned in the title of the manuscript. To show these critical points, this manuscript uses effectively purified proteins and steady-state kinetics.

Suggested revision:

This is a nicely written manuscript that addresses critical questions in translational control. The manuscript features some novel ideas in the field. The figures are aided by some effective illustrations of experimental designs and mechanisms. Supplemental material adds value, but does not contain core content. The work is of a high standard, well controlled, logically narrated, and significantly broadens our knowledge on regulation of gene expression.

The work could be very significantly bolstered by overexpressing eIF5, eIF2B-γ alone and in combination with eIF2B-ε, if doable. High copy eIF5 or 2B-γ alone should have no effect on the observed Gcd- phenotype, whereas 2B-γ plus ε should antagonize it like high copy IMT does.

---

## [Author Response]

*Suggested revision:*

*This is a nicely written manuscript that addresses critical questions in translational control. The manuscript features some novel ideas in the field. The figures are aided by some effective illustrations of experimental designs and mechanisms. Supplemental material adds value, but does not contain core content. The work is of a high standard, well controlled, logically narrated, and significantly broadens our knowledge on regulation of gene expression.*

*The work could be very significantly bolstered by overexpressing eIF5, eIF2B-γ alone and in combination with eIF2B-*ε*, if doable. High copy eIF5 or 2B-γ alone should have no effect on the observed Gcd- phenotype, whereas 2B-γ plus* ε *should antagonize it like high copy IMT does.*

As requested we have performed a new experiment with replaces the existing Figure 4. The new figure includes the requested comparisons and western blots to indicate which factors are over-expressed and that excess tRNA does not affect expression. As predicted by the reviewers we find that the growth defect on 3AT caused by over-expressing eIF5 is not suppressed by excess tRNA and eIF2B-γoverexpression has no phenotype to suppress. Co-overexpression of eIF2B- ε with γ behaves like eIF2B-ε alone over-expression. While the last result may not have been expected, we note that the levels of excess eIF2B-γ are not as high as excess eIF2B-ε in these experiments. Hence there likely remains significant excess free eIF2B-ε that likely contributes to the observed phenotype.